# Pooling strategies in V1 can account for the functional and structural diversity across species

**Victor Boutin** [ID]*, **Angelo Franciosini** [ID], **Frédéric Chavane** [ID], **Laurent U. Perrinet** [ID]

Inst. Neur. Timone, Aix-Marseille Univ, Marseille, France

* victor_boutin@brown.edu

## Abstract

Neurons in the primary visual cortex are selective to orientation with various degrees of selectivity to the spatial phase, from high selectivity in simple cells to low selectivity in complex cells. Various computational models have suggested a possible link between the presence of phase invariant cells and the existence of orientation maps in higher mammals' V1. These models, however, do not explain the emergence of complex cells in animals that do not show orientation maps. In this study, we build a theoretical model based on a convolutional network called Sparse Deep Predictive Coding (SDPC) and show that a single computational mechanism, pooling, allows the SDPC model to account for the emergence in V1 of complex cells with or without that of orientation maps, as observed in distinct species of mammals. In particular, we observed that pooling in the feature space is directly related to the orientation map formation while pooling in the retinotopic space is responsible for the emergence of a complex cells population. Introducing different forms of pooling in a predictive model of early visual processing as implemented in SDPC can therefore be viewed as a theoretical framework that explains the diversity of structural and functional phenomena observed in V1.

## Author summary

Cortical orientation maps are among the most fascinating structures observed in higher mammals' brains: In such retinotopic maps, preferred orientations in the cortical surface are clustered such that similar orientations activate neighboring cells, and orientation preference changes gradually along the cortical surface. However, the computational advantage brought by these structures remains unclear, as some species (rodents and lagomorphs) completely lack orientation maps. In this study, we introduce a computational model that links the presence of orientation maps to a class of nonlinear neurons called complex cells. In particular, we propose that the presence or absence orientation maps results from the diversity of strategies employed by different species to generate invariance to complex natural stimuli. These results have important applications for our understanding of how diverse biological organisms can achieve a given function (here low level-

**Data Availability Statement:** The data used to train the algorithm are publicly available at: https://cs.stanford.edu/~acoates/stl10/.

**Funding:** VB, AF, FC, LUP have received support from the French government under the Programme Investissements d'Avenir, Initiative d'Excellence d'Aix-Marseille Université via A*Midex (AMX-19-IET-004) and ANR (ANR-17-EURE-0029) funding. VB, AF, FC, LUP have used the ressources of the "Centre de Calcul Intensif d'Aix-Marseille" (ANR-10-EQPX-29-01) is acknowledged for granting access to its high performance computing resources. The funders had no role in study design, data collection and analysis, decision to publish, or preparation of the manuscript.

**Competing interests:** The authors have declared that no competing interests exist.

vision) and also for the elaboration of novel mechanisms in artificial neural network architectures such as convolution neural networks.

## Introduction

Cells in the primary visual cortex of higher mammals (V1) are sensitive to oriented localized visual patterns and these cells have classically been divided into two classes: simple and complex [1]. Simple cells show in particular a dynamic response to a drifting sinusoidal grating which is linearly modulated by the rectified temporal component of the sinusoidal grating: Simple cells are maximally activated when the stimulus matches a specific spatial phase inside their Receptive Field (RF) [1–3]. On the other hand, complex cells remain relatively invariant to the phase of the stimulus [1, 2, 4, 5]. As such, it is assumed that the primary visual cortex extracts elementary orientation features and produces scene representations that are both selective and invariant to the phase component [6]. Another remarkable property of the visual cortex is the hierarchical organization of the areas downstream to V1 along the ventral visual stream. Each of these areas is sensitive to increasingly complex features: simple edges for the primary visual cortex (V1), shape and textures for V4 [7], and specific objects in infero-temporal region (IT) [8]. These properties are in line with the challenging tasks of object recognition [9]: a stimulus is decomposed into simpler features throughout the hierarchical organization to build a representation that is specific enough to allow accurate recognition (selectivity) while being invariant to properties that do not affect the object identity (invariance) [10]. Historically, those properties of the visual cortex have been modeled with hierarchical networks which alternate layers of linear filters to describe simple cells with non-linear layers to account for complex cells invariance. In particular, Sakai and Tanaka [11] demonstrated that spatial pooling of simple cells units was necessary to reproduce complex cells behavior. Spatial pooling functions, exemplified by the sum of squared simple cells responses (i.e. energy pooling) or with a winner-take-all mechanism (i.e. max-pooling) [12], account for both spatial and phase invariance as it is observed in biological complex-cells.

Another phenomenon observed in the cortex of higher mammals is the possible presence of *orientation maps*. An orientation map is a functional structure of the selectivity of cells on the topography of the cortical surface for which neighboring cortical cells are preferentially tuned to similar orientations. Orientation preference varies smoothly but also shows local singularities, called pinwheels [13–15]. Both the formation mechanism [16–18] and the function [19–21] of such a topology have been thoroughly discussed in the literature. For example, [19] has proposed that orientation maps were optimal to ensure the uniform coverage of features over the visual space and [20] suggested that the presence of pinwheels could facilitate contour extraction. While some frameworks have successfully modeled orientation maps [22, 23], only a few of them have been proposed to make the link between complex cells and topographical organization. Hyvarinen *et al.* [24] observed that maximizing the sparseness of locally pooled simple-cells results in complex cells that were topographically organized in orientation maps. Similar results were obtained with a type of independent component analysis (ICA) [25]. Antolik *et al.* [26] managed to demonstrate that complex cells organize themselves in orientation maps if one includes a lateral inhibition mechanism. Further in that direction, clustering orientation preference within orientation maps could also facilitate the emergence of orientation-tuned complex receptive fields. Indeed, to keep orientation tuning while gaining invariance, complex receptive fields must be able to pool locally from all different phase selectivities within a set of similarly orientation tuned neurons. The emergence of orientation maps would

facilitate such an operation. However, there is no neurophysiological evidence of a systematic link between the emergence of complex cells and orientation maps. A topographic organization lacking an orientation map is commonly referred to as salt-and-pepper. For example, it is observed that there is no spatial structure for orientation preference in rodents (most notably mice, rats, and squirrels), although they do possess complex cells in V1 [14, 15]. The models listed above, cannot explain the presence of these complex cells in animals that do not present orientation maps in V1. This questions the actual links that can exist between orientation maps and complex cells. In particular, can the same framework be used to describe both salt-and-pepper and orientation maps structures while accounting for the emergence of complex cells?

In this article, we study the link between the emergence of topographical organization of orientation preference along with the emergence of complex cells by varying the pooling strategies between layers, using the Sparse Deep Predictive Coding (SDPC) as a model of V1 [27, 28]. The SDPC model integrates three key computational properties observed in the visual cortex: selectivity, invariance, and hierarchical architecture. First, selectivity in the SDPC is achieved with Sparse Coding (SC). Besides being justified by the efficient coding hypothesis [29], SC has successfully accounted for V1 simple cell's responses and Receptive Fields (RFs) [30, 31]. In addition, the SDPC leverages the Predictive Coding (PC) framework to describe the hierarchical relationship between the network's layers. PC suggests that every cortical area predicts at best the upstream sensory information. The mismatch between the prediction and the lower-level activity elicits a prediction error that is used to adjust the neural response until an equilibrium is reached [32–34]. The SDPC also includes convolutional layers that allow us to define distinct pooling functions acting in different subspaces: the spatial retinotopic space, and the feature space. In this work, we study the combined impact of these pooling strategies on both the emergence of complex cells and topographical maps. We demonstrate that pooling across the 2D *feature* space leads to a topographical organization similar to orientation maps, with the presence of phase maps but that does not account for the emergence of complex cells. In contrast, pooling across the *spatial* dimension leads to cells with complex-like behavior but without orientation maps. Only the combined pooling on both feature and spatial dimensions allows the common emergence of orientation maps, complex cells, and no phase maps. More generally, we argue that the combination of these different pooling strategies allows us to describe the structural and functional diversity across species. Then, we present the SDPC as a unifying theoretical model of the early visual cortex, which building principles can explain the link between topographical structures (retinotopic maps and orientation maps) and the receptive field structure (complex cells). The paper is organized as follows: First, we describe our model by detailing the SDPC as well as the pooling strategies. Next, we analyze the complex-like behavior and the topographical organization of the neurons for different pooling strategies. Then, we evaluate the impact of the network size on the pinwheels' density. Finally, we discuss our results and detail the implication of this work with respect to the state of the art.

## Results

### Brief description of Sparse Deep Predictive Coding (SDPC)

The Sparse Deep Predictive Coding (SDPC) framework solves a series of hierarchical inverse problems with sparsity constraints. A group of neurons $\gamma_i$ predicts by an optimization procedure the pooled activity from the previous cortical layer $p_{i-1}(\gamma_{i-1})$ and that was obtained through a set of synaptic weights $W_i$. Given a network with $N$ layers, we can define the

generative model [27, 28] as:

$$
\begin{cases}
x = W_1^T \gamma_1 + \epsilon_1, & \text{s.t. } \|\gamma_1\|_0 < \alpha_1 \text{ and } \gamma_1 > 0. \\
p_1(\gamma_1) = W_2^T \gamma_2 + \epsilon_2, & \text{s.t. } \|\gamma_2\|_0 < \alpha_2 \text{ and } \gamma_2 > 0. \\
\dots \\
p_{N-1}(\gamma_{N-1}) = W_N^T \gamma_N + \epsilon_N, & \text{s.t. } \|\gamma_N\|_0 < \alpha_N \text{ and } \gamma_N > 0.
\end{cases}
\tag{1}
$$

Where $x$ represents the input stimulus and $\gamma_i$ are the rate-based neural responses for layer $i$ (see Fig 1b). Sparsity constraints are introduced using the $\ell_0$ pseudo-norm which computes the number of active elements in each activity map $\gamma_i$. The $W_i$ matrices represent the network's weights (convolutional kernels), and $\epsilon_i$ is the prediction error associated to each layer. In Eq 1, $p_i$ denotes the pooling operator we use to model the complex cells. We have tested different type and combination of pooling functions based on max-pooling (see the section 'Pooling functions' for a detailed description). Henceforth, we use a 2 layer version of the SDPC to model V1 (i.e. $N = 2$ in Eq 1). To tighten the parallel with biology, we index all the first and second layer variables with the letter S and C to denote the simple and complex cell's layer, respectively.

One possibility to solve the generative problem defined in Eq 1 is by minimizing the following loss function [27]:

$$
F = \frac{1}{2}\|\epsilon_S\|_2^2 + \frac{1}{2}\|\epsilon_C\|_2^2 + \lambda_S\|\gamma_S\|_1 + \lambda_C\|\gamma_C\|_1 \text{ with } \epsilon_S = x - W_S^T\gamma_S \text{ and } \epsilon_C = p_S(\gamma_S) - W_C^T\gamma_C
\tag{2}
$$

In Eq 2, $\|\cdot\|_2$ and $\|\cdot\|_1$ denote the $\ell_2$ and $\ell_1$-norm, respectively. Note that the $\ell_0$ constraint in Eq 1 has been relaxed with a $\ell_1$-penalty term in Eq 2 as it leads to a more convenient optimization problem [35]. The minimization of $F$ in Eq 2 is performed using an alternation of an inference and a learning step. The **inference** step involves finding the optimal neural responses (i.e. $\gamma_i$) with a gradient descent on the loss function $F$ [36]:

$$
\gamma_S^{k+1} = \mathcal{T}_{\eta\lambda_S}^+ \left(\gamma_S^k - \eta\frac{\partial F}{\partial\gamma_S^k}\right) = \mathcal{T}_{\eta\lambda_S}^+ \left(\gamma_S^k + \eta W_S\ \epsilon_S - \eta\epsilon_C\ p_S^{-1}(\gamma_S^k)\right)
$$

$$
\gamma_C^{k+1} = \mathcal{T}_{\eta\lambda_C}^+ \left(\gamma_C^k - \eta\frac{\partial F}{\partial\gamma_C^k}\right) = \mathcal{T}_{\eta\lambda_C}^+ \left(\gamma_C^k + \eta W_C\ \epsilon_C\right)
\tag{3}
$$

Once the inference process has converged, the optimization of $W_S$ and $W_C$ is performed through the **learning** process:

$$
W_S^{k+1} = W_S^k - \omega\ \frac{\partial F}{\partial W_S^k} = W_S^k + \omega\ \gamma_S^T\ \left(x - W_S^T\gamma_S\right)
$$

$$
W_C^{k+1} = W_C^k - \omega\ \frac{\partial F}{\partial W_C^k} = W_C^k + \omega\ \gamma_C^T\ \left(p_S(\gamma_S) - W_C^T\gamma_C\right)
\tag{4}
$$

In Eqs 3 and 4, $\gamma_S^k$, $\gamma_C^k$, $W_S^k$ and $W_C^k$ denote the neural activity and the synaptic weights at the gradient step number $k$, respectively. $\eta$ is the step size of the inference process and $\omega$ is the step size of the learning. Additionally, $\lambda_S$ and $\lambda_C$ are the sparsity-inducing regularization parameters of the simple and complex cell's layer, respectively. $\mathcal{T}^+$ is the non-negative soft-thresholding operator (see Eq 7 in section 'Detailed description of SDPC' for a mathematical description of the soft-thresholding operator). In Eq 3, $p_i^{-1}$ denotes the approximation of the derivative of the max pooling function. Note that max-pooling is not differentiable but its derivative could be approximated by an unpooling function, i.e. an operator composed of a matrix filled with ones in the position of the local maximum elements and with zeros

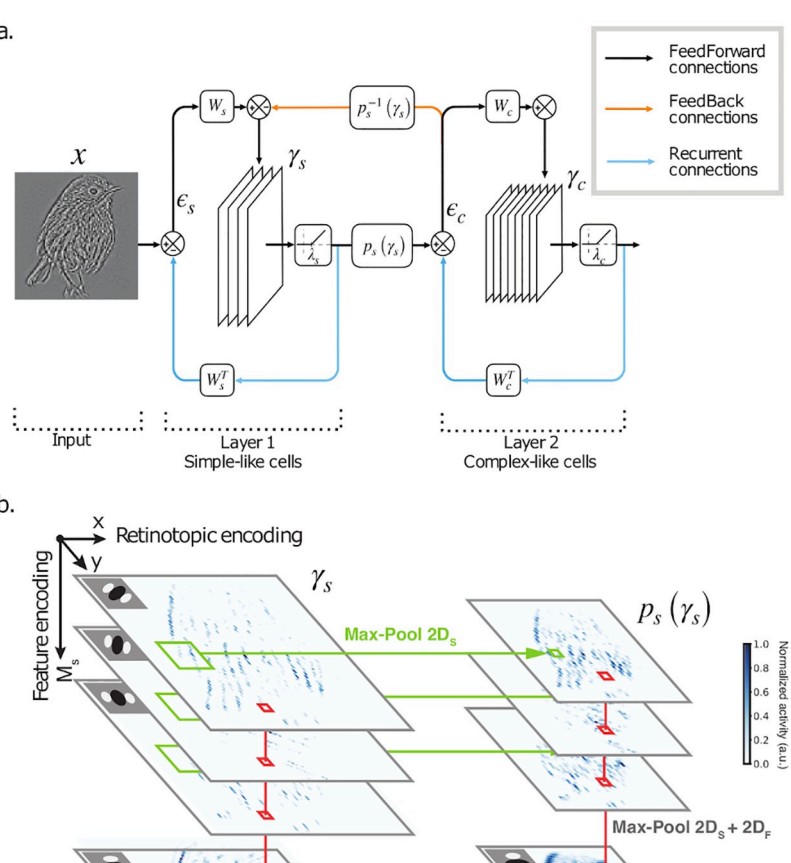

**Fig 1. The SDPC network. a**. Update scheme of the SDPC network used in this study: $x$ is the input image, $\gamma_s$ and $\gamma_c$ represent simple and complex cells response maps, respectively. $W_s$ and $W_c$ are convolutional kernels that encode for the RFs of the simple and complex cell layers, where each synaptic weight matrix is composed of $M_s$ and $M_c$ neurons (kernels) respectively. $p_s(\gamma_s)$ is the pooling function used to generate position and feature invariance in the response of the second layer. Feed-forward connections carry information on the prediction errors ($\epsilon_s$ and $\epsilon_c$) that are used to refine the neural activities. Feedback information is carried through the unpooling function $p_s^{-1}(\gamma_s)$, that approximates the derivative of the pooling function. The circles with arrows inside are error nodes such that the output signal is equal to the difference of the input signal. **b**. Here, we show a representation of $\gamma_s$ and $p(\gamma_s)$. Each pixel represents a model neuron and the color code indicates the amplitude of the neural response (lighter for no response and darker blue for the maximal response, here normalized to 1). In this figure, we illustrate three possible outputs $p_s(\gamma_s)$, used to generate different network structures: MaxPool $2D_S$ selects the maximum activity over spatial (retinotopic) positions, in each plane independently; MaxPool $2D_F$ acts in the feature space by selecting the maximum activity across planes in $\gamma_s$. When the network integrates the two pooling functions in sequence, we refer to them as a unique max-pooling operator called MaxPool $2D_S + 2D_F$.

everywhere else. Fig 1a shows the update scheme of the SDPC network used in this study. In this study, we aim to model simple and complex cells in V1.

## Pooling functions

All pooling functions are based on max-pooling: a function that selects the maximum response within a group of cells (here, $\gamma_{s1}$ and $\gamma_{s1}$ for an example):

$$p(\gamma_{s1}, \gamma_{s2}) = \max(\gamma_{s1}, \gamma_{s2}) \tag{5}$$

The max-pooling performs a winner-takes-all computation rather than calculating the energy of the firing of a pool of neurons [11]. We decided to use max-pooling as a way of generating invariance for two reasons: First, the winner-take-all mechanism can account for numerous nonlinear responses in the sensory cortex, and it is regarded as one of the fundamental components of sensory processing in the brain [10, 11, 37, 38] (see section 'About the bio-plausibility of SDPC' for more details on the bio-plausibility of max-pooling); Second, the local competition enforced by max-pooling is echoing the global competition mechanism introduced with Sparse Coding (SC). Indeed, while max-pooling performs a winner-takes-all computation in cells that belong to the same neighborhood, SC forces all neurons in a given layer to compete with each other to best predict the input. Using both SC and max-pooling to model respectively simple and complex cells allows us to reduce the core computation of V1 to a single mathematical competition mechanism.

Given a neural response map, $\gamma_s$, max-pooling can be seen as a nonlinear convolution whose output, $p_s(\gamma_s)$, is the maximum value in each pooling region (see Fig 1b). Just as convolution, max-pooling is defined by its kernel size and stride. In this study, by varying these parameters, we introduce three types of pooling functions:

- MaxPool $2D_S$ acts in the spatial dimension, independently for each feature. In this study, we use a pooling kernel of $2 \times 2$ neurons with a stride of 2.

- MaxPool $1D_F$ selects the maximum across a neighborhood of adjacent planes of $\gamma_s$ along with a $1D$ circular space. This function acts in the feature space, leaving the spatial encoding unchanged. For this function, we used a linear kernel of size 4 and a stride equal to 1.

- MaxPool $2D_F$ arranges a feature space composed of $M$ neurons on a grid of $\sqrt{M} \times \sqrt{M}$ grid, for each spatial location. Then it selects the maximum activity across $2D$ pooling regions, with a size of $2 \times 2$ neurons with a stride of 1. This pooling function only acts in the feature space, just as the one we defined above.

Since the pooling functions operate in two different sub-spaces, we can apply them in sequence to combine the effect of the two strategies (see Fig 1b). We call these functions Max-Pool $2D_S + 1D_F$ and MaxPool $2D_S + 2D_F$.

## Pooling in a predictive coding network

Besides reaching a stable point in terms of the loss function (see Eq 2), the network efficiently developed edge-like Receptive Fields (RFs) in the first layer (see Fig 1b and S1 Fig top left panel) and second layer (S1 Fig bottom left panel). Note that even if the convergence of the SDPC is guaranteed due to the convexity of the loss function, nothing prevents the SDPC to converge towards a trivial solution. This first observation offers then a sanity check and confirms that the SDPC is converging towards a meaningful solution. This result holds for the different network sizes that we tested (36, 49, 64, 81, 100 and 121 neurons for each layer) and the different combinations of pooling functions: MaxPool $2D_S$, MaxPool $2D_F$, MaxPool $2D_S + 1D_F$, and MaxPool $2D_S + 2D_F$. Note that inferring the shape of the second layer RFs is not trivial, because the pooling function $p_s(\gamma_s)$ makes it challenging to project the learned filters back into the space of the input image (see [28] for details). In S1 Fig, we show $V_c$, a linear approximation for the second layer kernels (see section 'Log-Gabor fitting' for more detail on this linear approximation). From these linear projections, we can see that the $W_c$' RFs also have the shape of localized edge-like filters (see S1 Fig bottom left panel). This is interesting to observe that the second layer RFs are similar to the first layer RF even if the second layer response is strongly

non-linear (due to the max-pooling operation). In the following section, we evaluate the difference between the first and second layer response.

## Phase invariance and complex behavior

In Fig 2 we show some typical responses of complex and simple cells when varying both orientation and phase. While all of the first layer neurons are sensitive to both orientation and phase (i.e. simple cells), we observe that some of the second layer neurons are invariant to phase variation (i.e. complex cells, see Fig 2). To quantify phase invariance at the population level we use the modulation ratio, denoted $\frac{F_1}{F_0}$ and introduced by [4] (see section 'Modulation ratio and complex behavior' for mathematical details on this index). A cell is classified as simple if $\frac{F_1}{F_0} > 1$ and as complex if $\frac{F_1}{F_0} < 1$. We test the phase invariant behavior of the network when trained using 4 different settings corresponding to the following pooling strategies: $p_s$ = Max-Pool $2D_S$, $p_s$ = MaxPool $2D_F$, $p_s$ = MaxPool $2D_S + 1D_F$ and $p_s$ = MaxPool $2D_S + 2D_F$. In all these settings, the distribution of $\frac{F_1}{F_0}$ shows that the first layer develops exclusively simple-like neurons, while the second layer shows a distribution that depends on the pooling strategy (Fig 3, fourth column). When spatial pooling and circular feature pooling are combined together (for $p_s$ = MaxPool $2D_S + 1D_F$), we observe a sharp high peak of $\frac{F_1}{F_0}$ in 0 in the second layer (see Fig 3c). In contrast, the $\frac{F_1}{F_0}$ distribution is much broader when the spatial pooling is combined

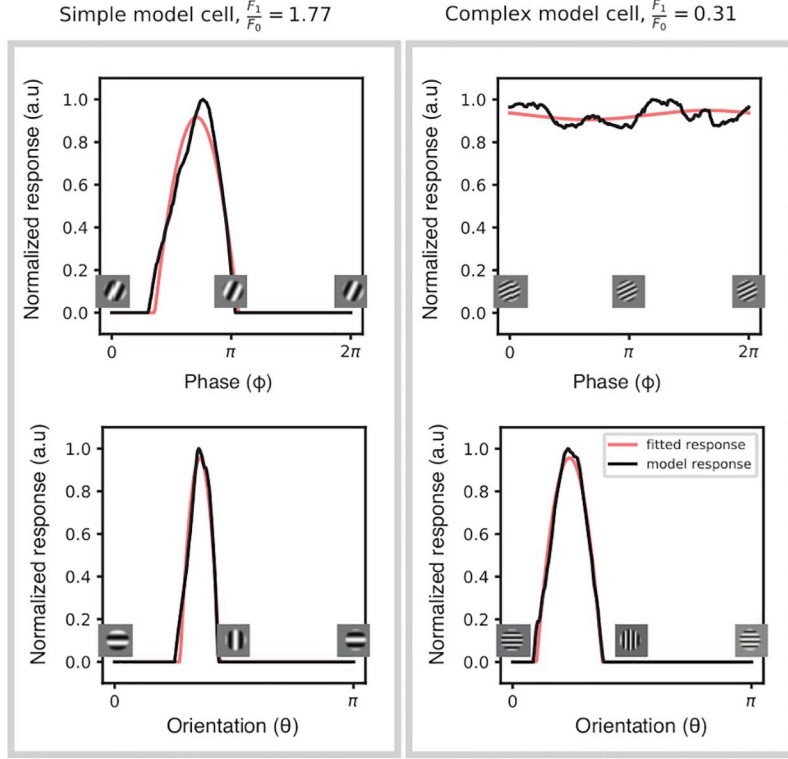

**Fig 2. Response of simple-like and complex-like model neurons.** Example of two model neurons exhibiting a simple (**left**) and complex (**right**) behavior. The black lines indicate the model neuron's response when its receptive field contains a drifting (**top**) or rotating (**bottom**) stimulus; the red lines indicate the response modeled according to a half-rectified sinusoidal model as in [39]. The modulation ratio, $\frac{F_1}{F_0}$, is greater than 1 for simple neurons that are tuned to a specific phase. A complex response, that is partially or completely independent to phase, is quantified by a low modulation ratio (less than 1). Importantly, both neurons remain tuned to orientation.

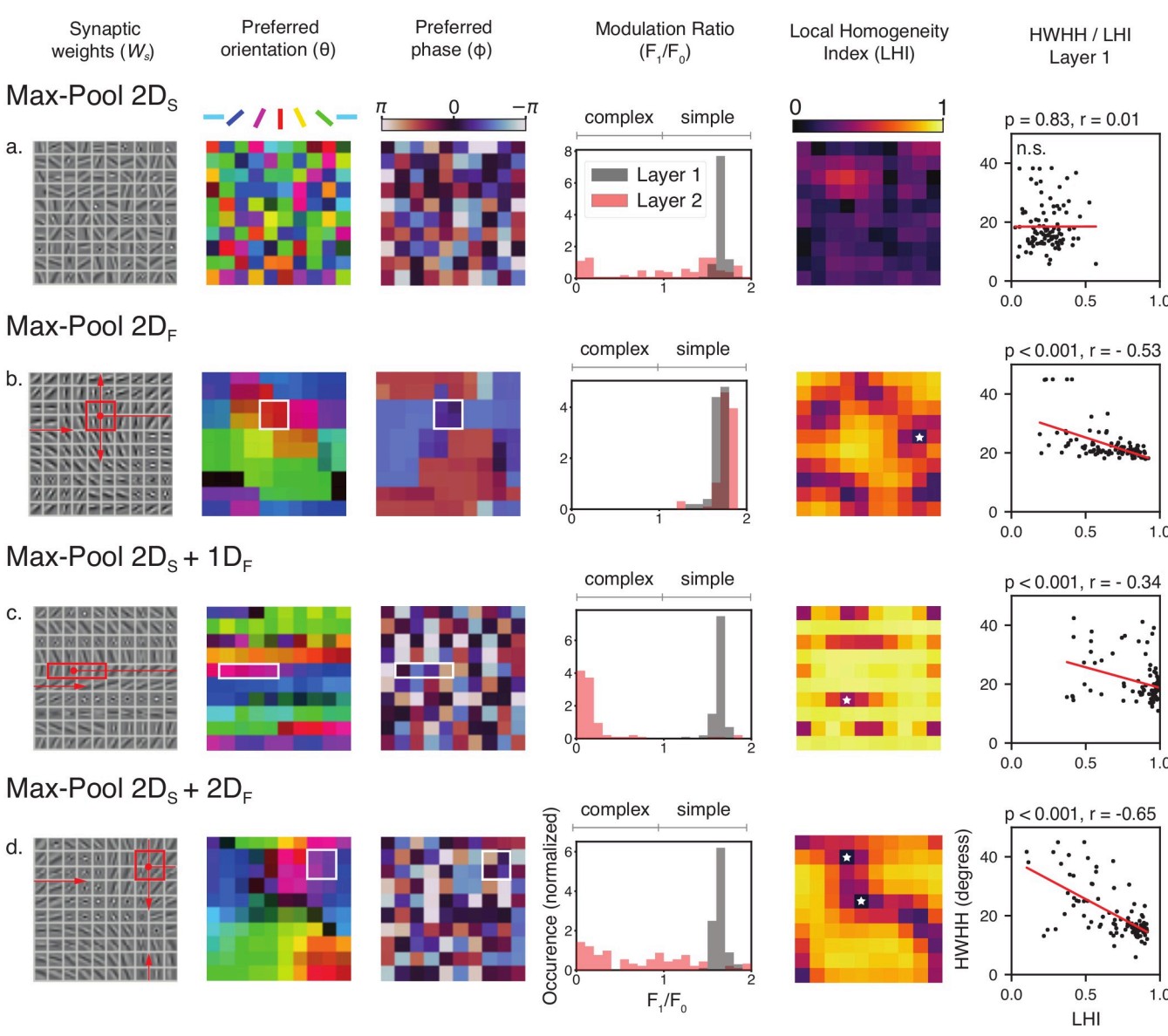

**Fig 3. Emergence of orientation maps in the first layer and complex response in the second layer.** We show the different properties learned by our model depending on which pooling functions we used (**a**—**d**). All networks showed here are trained with $M_s = 100$ and $M_c = 100$. **First column**. Representation of the simple cells Receptive Fields (i.e. the synaptic weights of the first layer $W_s$). The red rectangles for (**b**—**d**) indicate the size of the feature pooling, the red arrows illustrate the circular (**c**) or toroidal (**b** and **d**) structure of the feature space. **Second column**. Orientation preference of each neuron represented by the filters in $W_s$. The luminance of the map is modulated by the orientation selectivity (HWHH) of each neuron. **Third column**. Phase preference of each neuron represented by the filters in $W_s$. **Fourth column**. Distribution of the modulation ratio ($\frac{F_1}{F_0}$) for the first and second layers of the network. **Fifth column**. Local homogeneity index (LHI) associated with each element is $W_s$. White stars indicate the position of pinwheels. **Sixth column**. Linear relationship between the LHI and the orientation selectivity, measured by the half-width at half-height (HWHH) of the response of cells from the first layer.

with 2D feature pooling (for $p_s$ = MaxPool $2D_S$ + $2D_F$, see Fig 3d). For these 2 previous settings, the second layer neurons are strongly invariant to phase (high peak at 0 in Fig 3c and 3d). When $p_s$ = MaxPool $2D_F$, our model does not develop complex cells, i.e. the $\frac{F_1}{F_0}$ is high for both layers (see Fig 3b). When we impose only a spatial pooling (i.e. $p_s$ = MaxPool $2D_S$), the second layer of our network exhibits as many simple cells as complex cells (see Fig 3a).

## Learning topographic orientation maps in the SDPC network

Networks trained using pooling functions that act in the feature space (MaxPool $2D_F$, MaxPool $2D_S + 2D_F$, and MaxPool $2D_S + 1D_F$) showed the emergence of differential topographic structures for $W_s$ and, by consequence, $\gamma_s$. Interestingly, as we introduce pooling in the feature space (see section 'Pooling functions' for more details on pooling functions), the arrangement of filters on $W_s$ converges to form a topographic orientation map where neighboring neurons become tuned to similar orientations (see Fig 3b–3d, second column). To compare quantitatively the orientation maps learned by our model with the ones observed in neurophysiological experiments, we used the local homogeneity index (LHI), introduced by [40] (for mathematical details on the LHI index see section 'Local homogeneity index (LHI)'). We observed in the fifth column of Fig 3b–3d that the orientation maps developed by the SDPC contain regions where orientation preference varies smoothly (high LHI) combined with local discontinuities (low LHI), in analogy with pinwheels observed in higher mammals [13–15]. One may wonder if the existence of these topographic maps has functional implications for the properties of neurons from the first layer. Indeed, electrophysiological experiments have shown a clear link between a neuron's position in the cortical map and its tuning properties [20, 40] (but see [13]). Specifically, neurons located near pinwheels tend to have a broader orientation tuning, while neurons located in iso-orientation regions display a narrower, more selective tuning. We evaluated the relationship between the LHI of single neurons in the first layer and their orientation tuning, evaluated as the half-width at half-height (HWHH) of their response to a rotating grating. We found a linear relationship similar to one of [40] and [20]. This relationship is valid for all networks with a $2D$ topographic map ($p_s$ = MaxPool $2D_S + 2D_F$) and $M_s \geq 64$ ($p < 0.01$). The linear relationship is significant ($p < 0.01$) for few configurations of $p_s$ = MaxPool $2D_F$ and $p_s$ = MaxPool $2D_S + 1D_F$, with no clear dependence on $M_s$.

Interestingly, the networks trained with a combination of spatial and feature pooling (MaxPool $2D_S + 1D_F$ and MaxPool $2D_S + 2D_F$) appear to develop a topographic map when we look at the preferred orientations $\theta$ of neurons in the first layer (Fig 3c and 3d, second column). However, this structure is not present if we look at the preferred phase map $\phi$, where there appears to be no particular structure (Fig 3c and 3d, third column), in line with the known neurophysiology [41]. This is not true in the case of MaxPool $2D_F$, where both maps appear to be organized in groups of cells sensitive to similar orientations and phases, in contrast with neurophysiological evidence (see Fig 3b, second and third column). To further analyze the orientation map structures, we used the LHI. The LHI quantifies the number of pinwheels generated by our model in different tested conditions (for mathematical details on the pinwheels density calculation see section 'Local homogeneity index (LHI)'). We found that the tested pooling functions, MaxPool $2D_F$, MaxPool $2D_S + 2D_F$, and MaxPool $2D_S + 1D_F$, generated a comparable number of pinwheels centers even for networks about 4 times bigger than the smallest network tested: $M_s = 36$ and $M_s = 121$ (see Fig 4). Thus, the number of pinwheels singularities remarkably does not depend on the size of the first layer $M_s$. A similar effect has been reported across many species [42].

## Invariance to phase vs. invariance to orientation

We want to further explore why the complex cells developed by the second layer of our model show invariance to the stimulus phase while remaining tuned to orientation (see Fig 5). A highly nonlinear network could respond unspecifically to a broad set of different stimuli. Invariance to multiple properties of the stimuli (e.g. both to phase and orientation) would be in contradiction with neurophysiological experiments. In fact, it is well established that complex cells exhibit phase invariance but show orientation tuning similar to simple cells [4, 5]. To

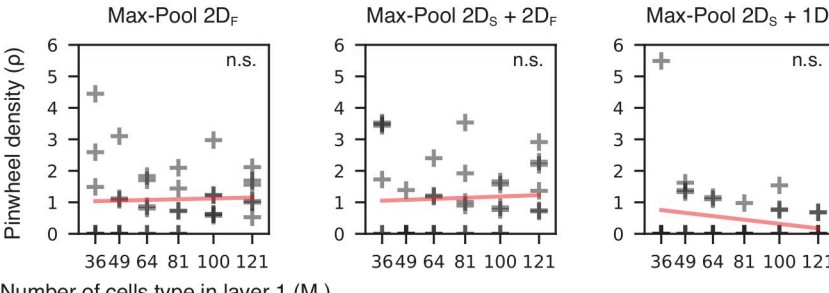

**Fig 4. Pinwheel density does not depend on network size.** Here we show that there is no relationship between the tested first layer's network size ($M_s$) and the density of pinwheels ($\rho$). Pinwheel density is defined as the average number of pinwheels per area unit (see section 'Local homogeneity index (LHI)'). For all networks showing orientation maps, we found no significant trend between the size of the map, $M_s$, and the pinwheel density, in line with neurophysiological observations.

assess whether the network's response is orientation-selective or not, we tested the network's invariance property to both phase and orientation of gratings. We used drifting gratings as stimuli that change phase over time. Conversely, we used rotating gratings that change orientation, to modulate the response of orientation-selective neurons. This was performed for each cell of the second layer of our network at its optimal spatial frequency (see section 'Drifting grating vs. Rotating grating'). For quantification purposes, we define R as the percentage of cells that show a complex-like response for phase, or that are unselective to orientation ($\frac{F_1}{F_0} < 1$). We report the results relative to the second layer of the networks, as in all the tested conditions, the first layer of the network exclusively exhibited simple-like responses ($R_\phi = 0$, see Fig 3, fourth column). In Fig 5, the black and green lines $R_\phi$ and $R_\theta$ correspond to the dependence on phase (drifting) and orientation (rotating), respectively. We analyze these results as a function of the number of channels in the first layer, $M_s$ to evaluate the impact of the network structure on the second layer's responses. For $p_s$ = MaxPool $2D_S$, the networks showed high $R_\phi$ (up to 60%) and low $R_\theta$ (at most 20%), independently to $M_s$. This suggests that the SDPC network efficiently develops phase-invariant responses while maintaining a relatively narrow tuning to the stimulus's orientation (see Fig 5, top-right). When $p_s$ = MaxPool $2D_S + 1D_F$, that is, a spatial pooling and a feature pooling organized on a linear structure, $R_\phi$ is much higher (up to 100%, see Fig 5, bottom-right) and a smaller $R_\theta$, also independent on $M_s$; this can be explained by the combined action of pooling across spatial locations and across features with same orientation but different phases (see Fig 3). Interestingly, only when we impose $p_s$ = MaxPool $2D_S + 2D_F$, the network's behavior appears to vary as a function of $M_s$ (see Fig 5, bottom-left). For low dimensions (up to $M_s = 64$), the tested networks show a high fraction of complex-like cells. For $M_s > 64$, $R_\phi$ decreases as $M_s$ increases (see S1 Text for statistical tests). The same effect can be observed for $R_\theta$, which appears to be high for $M_s = 36$, and then decreases with increasing $M_s$.

## Discussion

In previous work [28], we introduced the SDPC algorithm, and we used it to model local interactions in the early visual cortex (V1/V2). We showed SDPC can predict how strong intra-cortical feedback connectivity (from V2 to V1) shapes the response of neurons in V1 according to the Gestalt principle of good continuation. In this article, we have extended the SDPC with different pooling operations such that it is now including this non-linear computation. Our 2-layer SDPC model of V1 allows us to test different types of pooling operation: MaxPool $2D_S$,

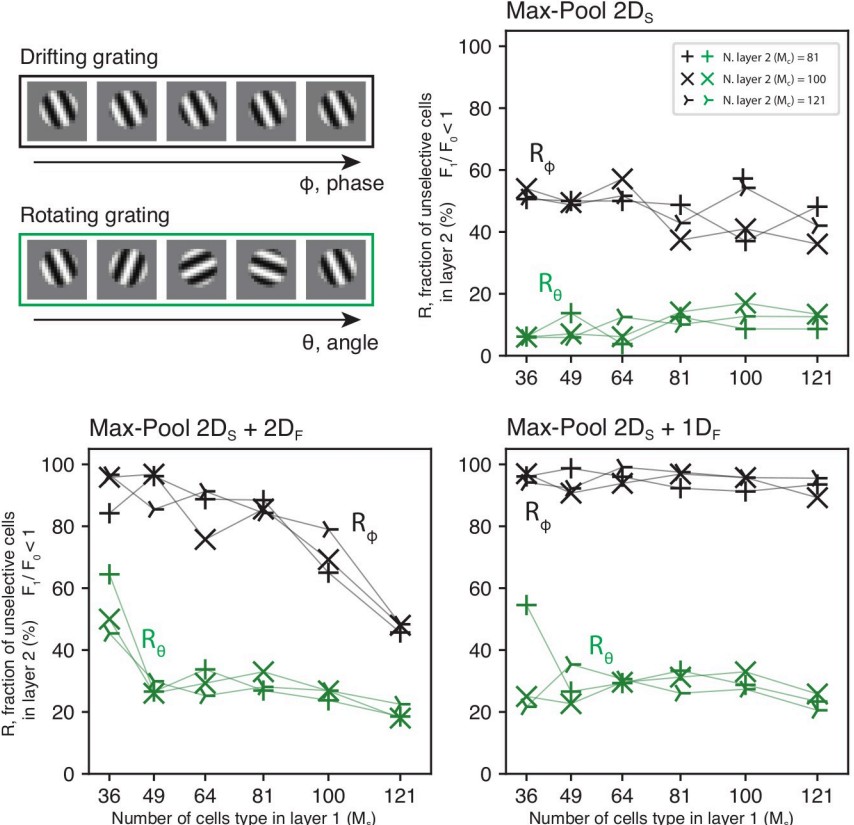

**Fig 5. Complex cells population analysis.** Here we show how the type of pooling function used and the number of cells type present in the first and second layers of the network ($M_s$ and $M_c$, respectively) affect the complex cells population's properties developed by the model. **Top-left**. We tested the network's second layer invariance to phase (drifting) and orientation (rotation) in sinusoidal gratings (see section 'Drifting grating vs. Rotating grating'). The ratios of cells $R_\phi$ (complex) and $R_\theta$ (orientation invariant), are defined as the percentage for which $\frac{F_1}{F_0} < 1$ in the second layer of the network. While the network's second layer shows a strong invariance to a drifting grating (black lines), the same effect is not present for the rotating grating (green lines). This result indicates that the network's invariant response is specific to the stimulus's phase and not to its orientation. Overall $R_\phi$ is much higher when the max-pooling acts also in the feature space ($p_s = \text{MaxPool } 2D_S + 1D_F$ and $p_s = \text{MaxPool } 2D_S + 2D_F$). Interestingly, for $p_s = \text{MaxPool } 2D_S + 2D_F$, the number of complex-like cells seems to depend on the size of the network's first layer $M_s$.

MaxPool $1D_F$ and MaxPool $2D_F$ and to assess their impact on the cell's response and on the emergence of an orientation map.

## Summary of results

We have shown that when we introduce a 2D spatial pooling only (i.e. MaxPool $2D_S$), our model efficiently develops complex cells, but we do not observe the emergence of an orientation map (see Figs 3a and 5). In the case of MaxPool $2D_F$, the network only pools in the feature space. In this condition, the model learns a topographic map in the first layer for both orientation and phase (Fig 3b) but no complex cells are present in the second layer ($R_\phi \approx 0\%$). For MaxPool $2D_S + 1D_F$, the network pools in both retinotopic and feature space (circular topology). In this case, the first layer of the network develops a topographic structure for orientation preference but not for phase (Fig 3c). The second layer presents a high fraction of complex cells (up to $R_\phi = 100\%$) in all tested conditions irrespective of $M_s$. For MaxPool $2D_S + 2D_F$, the network pools in both retinotopic and feature space (toroidal topology). Similar to the

previous case, the network develops, in the first layer, a topographic map for orientation but not for phase (Fig 3d). In this case, however, $R_\phi$ appears to depend on $M_s$, with smaller dimensions for the first layer (low $M_s$) appearing to produce more complex cells (Fig 5). Consequently, the different combinations of pooling functions, allow the SDPC to account for a high diversity of cells types and cortical maps. In the next section, we interpret our computational findings in light of current neuroscientific knowledge and link our work with other state-of-the-art models to discuss the bio-plausibility of the SDPC model.

## SDPC models the diversity of V1 cells' type and topological maps across species

We observed that when the pooling function used in the model acts in the feature space (MaxPool $2D_F$, MaxPool $2D_S + 1D_F$, and MaxPool $2D_S + 2D_F$), the SDPC model develops an orientation map in the first layer. These orientation maps show quantitatively strong similarities with those observed in higher mammals: First, we found that a linear relationship exists between the local homogeneity index (LHI) of a model neuron (i.e. its position relative to pinwheels in the map) and its orientation tuning as observed in neurophysiological studies [20, 40]. In particular, neurons near pinwheels (low LHI) have a broader tuning than those in iso-oriented regions (see section 'Learning topographic orientation maps in the SDPC network'). Second, we found the number of pinwheels for each of the conditions listed above to be independent of the network first layer's size, $M_s$ (Fig 4). To sum up, our model can predict, without any supervision mechanism, a key property of cortical orientation maps across different species such as carnivores and primates, that is, the emergence of pinwheels and orientation domains with a constant pinwheel density even for a large diversity of V1 sizes [14, 18, 42].

According to our model, complex cells emerge by pooling in the retinotopic space, even in absence of orientation maps, thanks to hierarchical pooling in V1. Pooling across retinotopic positions (MaxPool $2D_S$) can explain, by itself, the emergence of complex cells by enforcing **position invariance**, a fundamental mechanism observed in complex cells [43, 44]. In this case, the absence of cortical orientation maps suggests that the emergence of complex cells depends solely on pooling across different positions in the retinotopic space. This type of network could be the one preferably implemented in animals that do not exhibit orientation maps [14, 15]. This is in line with the results from [45], showing that orientation selectivity can emerge even in networks that connect locally neurons with a random preferred orientation, as in a salt-and-pepper map; in analogy with the random maps that the network produces in the case of MaxPool $2D_S$ [45, 46]. On the other hand, in the case of MaxPool $2D_S + 1D_F$ and MaxPool $2D_S + 2D_F$, our model can generate **feature invariance** by pooling in the feature space. As a consequence, the model converges to a configuration where complex cells can be generated by pooling across the same orientations and different phases (Fig 3c and 3d).

We have shown that networks that develop topographic maps, on top of classical spatial pooling, exhibit more complex-like cells and, in general, more phase invariant response (see section 'Phase invariance and complex behavior' and Fig 3c and 3d). These findings suggest that rodents should show a lower fraction of complex cells in V1, compared with other mammals. This prediction is, indeed, in line with neurophysiological findings in mouse [47, 48] and the rabbit [49], although squirrels (that are highly visual rodents) show a fraction of complex cells comparable to other mammals [46]. The most remarkable fact is that these results are obtained solely by changing a single computational mechanism, that is the type of nonlinear pooling used, and by learning neural responses directly from natural images.

### Predictive Coding as a mechanism for orientation map formation

In SDPC, the orientation maps are naturally emerging thanks to the combined action of prediction error minimization and pooling. We have observed that there is no orientation map in the first layer when we remove the feedback connection (i.e. the orange connection in Fig 1a). This observation suggests that the feedback from the second layer mediates the formation of cortical maps in the first layer. Interestingly, a similar relationship between simple and complex cells has been suggested by Kayser & Miller [50]. We can then hypothesize that the orientation map (in the first layer) represents the best possible organization to minimize the prediction error with a representation (in the second layer) that is locally feature-invariant (because of the pooling in the feature space).

### Relation to the state-of-the-art

One of the first models to account for both complex cells and orientation maps is the Topographic ICA from Hyvarinen *et al.* [25]. The topographic ICA describes the topology in V1 as a quantification of the residual dependence between the components of an ICA (the closer the features in the topological space, the more dependent they are). In addition, Hyvarinen *et al* have also observed orientation maps when maximizing the sparsity of a feedforward network with an energy pooling layer [24]. In contrast, the topology is naturally emerging from the combined action of feedback connection and pooling in the SDPC (see section Predictive Coding as a mechanism for orientation map formation). To the best of our knowledge, SDPC is the first model that links the emergence of the orientation map with the predictive coding framework. Another difference is that SDPC is convolutional while [24] and [25] have fully-connected neural layers. Fully-connected architectures do not disentangle the role of retinotopy from the one of orientation map in building complex cell responses, as the two structures are merged in the same map. The SDPC being convolutional, the feature space is by construction dissociated from the retinotopic space. This particular property allows the SDPC to describe not only the primate and carnivore orientation maps, but also the salt-and-pepper configuration observed in rodents (see section SDPC models the diversity of V1 cells' type and topological maps across species).

Other frameworks have been proposed to compute the optimal topography of features in natural image representation. For example, [51] propose a model based on strong dimension reduction (using PCA) that learns to ignore fine-grained structure from the signals of simple cells and discover a linear pooling of correlated units. Interestingly, [22] adopted a similar approach in which pooling emerges to model the geometry of the manifold of a sparse code. Different from the SDPC, these models have not studied the combination of complex cells with the emergence of different types of topographical maps (orientation map and salt and pepper).

### About the bio-plausibility of SDPC

In this article, we have modeled complex cells using the max-pooling function that relies on the *max* operation. As demonstrated by [38], a cortical circuitry could implement a *max*-like operation. Interestingly, the proposed canonical circuit could also be used to model energy pooling.

The SDPC model is relying on convolution operation. Convolutions assume that the synaptic weights are repeated across the image to tile the whole visual field. This weight-sharing mechanism is unlikely to be implemented in biology. However, the same retinotopic architecture can be achieved with local untied connections to keep the same structure without replicating the set of weights at each spatial position. In particular, [52] showed that such a model

converges to a network similar to a convolutional network. These results suggest that convolutions can be regarded as a good model of retinotopic processing, even if it is not strictly biologically plausible.

The SDPC satisfies computational constraints that are thought to occur in the brain. As illustrated by Eq 3, all computations involved in the neural response update are local. Indeed, the new state of the neural population (i.e. $\gamma_i^{k+1}$) only depends on its previous state (i.e. $\gamma_i^k$), the states of the adjacent layers ($\gamma_{i-1}^k$ and $\gamma_{i+1}^k$) and the associated synaptic weights ($W_i$ and $W_{i+1}$). In addition, one can observe that the weight update equation (see Eq 4) is a product of monotonically increasing functions of pre-synaptic ($\gamma_{i-1}$) and post-synaptic activity ($\gamma_i$). It could then be interpreted as an Hebbian rule [53] that have strong grounding in biology.

## Concluding remarks

In this study, we have shown that SDPC including different variations of max-pooling could be regarded as an unsupervised and simple computational framework to model the diversity of mammals' V1. It provides a plausible explanation for the emergence of complex cells in different types of topographical structures (salt-and-pepper and orientation maps) as observed in different species. One interesting perspective would be to extend to SDPC such that the pooling function can be learned rather than being fixed. To do so, one might leverage the more general parametrization of the pooling function that was proposed in [38]. Such a formalization would allow the SDPC to learn a continuum of pooling functions spanning from energy pooling to max-pooling. In addition, the SDPC allows us to analyze the effect of the feedback. In previous work, we have shown that more feedback strength in the SDPC, 1—reshapes neural organization to improve contour integration [28], 2—modulates the shape of the V1 RFs [27] and 3—improves generalization abilities [54]. Further work could then be conducted to assess the impact of feedback strength on the topology of features in the first layer.

## Methods

### Detailed description of SDPC

In the section 'Brief description of Sparse Deep Predictive Coding (SDPC)', we have described the generative problem solved by the SDPC model (see Eq 1). For the sake of concision, in Eq 2, we have given the loss function only for a 2-layered SDPC network. Here we showcase a more general formulation of this loss function, applicable to a $N$-layered network:

$$F = \frac{1}{2}\sum_{i=1}^{N}\|\epsilon_i\|_2^2 + \lambda_i\|\gamma_i\|_1 \quad \text{s.t. } \epsilon_1 = \ x - W_1^T\gamma_1$$

$$\text{and } \forall i \in [2, N], \ \epsilon_i = p_{i-1}(\gamma_{i-1}) - W_i^T\gamma_i \tag{6}$$

In this equation, $\epsilon_i$ encodes for the prediction error, $\gamma_i$ represents the neural response at the layer $i$, and $W_i$ denotes the synaptic weights (which has a convolutional structure) between the layer $i - 1$ and $i$. The $\lambda_i$ coefficient controls the strength of the sparse regularization constraint on the neural response $\gamma_i$. Convolution is an efficient mathematical framework to model retinotopic activity in the visual system: synaptic weights are repeated across the image to tile the whole input image. This brings the advantage of having a model that is translation invariant: the synaptic weights (also called features or kernels) encoded by $W$ are translated at each position of the input image (for more details on the bio-plausibility of the convolution, see section 'About the bio-plausibility of SDPC'). The resulting vector $\gamma$ can thus be viewed as a neural response map, whose pixels encode for neurons sensitive to a specific feature and at a specific

position in the input image, in perfect analogy with a retinotopic map (see Fig 1b). The other parameters of this convolution are the kernel size and the stride, as detailed in section 'Training'. The minimization of Eq 6 is performed using an alternation of **inference** (see Eq 3) and **learning** (see Eq 4) steps. The soft-thresholding operator $\mathcal{T}^+$ in Eq 3 is enforcing that the majority of neurons from $\gamma_i$ are inactive (it thus has a sparsifying effect). The mathematical definition of the soft thresholding operator is:

$$\mathcal{T}^+_{\eta\lambda_i}(x) = \begin{cases} x - \eta\lambda_i, & \text{if } x - \eta\lambda_i > 0. \\ 0, & \text{otherwise.} \end{cases} \tag{7}$$

## Training

We built different SDPC networks to model simple and complex cells in V1. Each network has convolutions with kernel sizes of $7 \times 7$ pixels for the first layer ($W_s$), and $4 \times 4$ for the second layer ($W_c$). For both layers, we used a stride of 1. We tested the network with 4 different pooling functions: MaxPool $2D_S$, MaxPool $2D_F$, MaxPool $2D_S + 1D_F$ and MaxPool $2D_S + 2D_F$. Importantly, we introduced a zero padding before the pooling to make sure that first and second layer cells have the same receptive field size with respect to the input image ($14 \times 14$ pixels). Each network was also trained with different neural populations sizes: with $M_s$ and $M_c$ equal to 36, 49, 64, 81, 100 and 121 neurons for the first and second layer respectively, leaving us with 36 networks for each tested pooling function. The kernel values $W_s$ and $W_c$ were initialized as random noise and were normalized during training such that the energy (Euclidean norm) of each kernel was set to 1. All the networks were trained using grayscale natural images from the STL-10 dataset ($96 \times 96$ pixels per image) [55] for 9 epochs (28125 iterations with mini-batches of 32 images). The input images were pre-processed with a whitening filter similar to the one used in [30] to model retinal processing. Each image was then bounded between the values $-1$ and 1. The synaptic weight $W_s$ and $W_c$ were updated using stochastic gradient descent with a learning rate, $\omega$, of 0.01 and a momentum, $\beta$, of 0.9. Additionally, the sparsity parameters $\lambda_s$ and $\lambda_c$ were gradually incremented during training up to a value of 0.1.

## Modulation ratio and complex behavior

In this study, in order to quantify the simple and complex behavior of model neurons in the SDPC, we use the classical $\frac{F_1}{F_0}$ measure, also known as the modulation ratio [4]. Given the response of a neuron to a grating drifting at temporal frequency $f$, the $\frac{F_1}{F_0}$ measure represents the ratio between the first harmonic of the response ($f = F_1$) and the mean spiking rate $F_0$. Intuitively, a high modulation ratio (between 1 and 2) indicates that the cell is sharply tuned to a specific phase and it is thus regarded as simple. If a cell shows a low modulation ratio (between 0 and 1) it is then regarded as complex, showing a broad tuning to the stimulus' phase. For a neural population containing simple and complex cells, the distribution of the modulation ratios will appear to be bi-modal, with two peaks roughly in the regions described above. In [39], authors showed that the modulation ratio can be derived analytically from a half-rectified model of the a neuron's response to a drifting grating. In particular, $\frac{F_1}{F_0}$ depends non-linearly on $\chi$ defined as:

$$\chi = \frac{V_{th} - V_{mean}}{|A|} \tag{8}$$

Where $V_{mean}$ and $V_{th}$ represent respectively the mean membrane potential and the threshold

for spiking generation and $A$ is the maximal amplitude of the modulation. Here we use the simplified rate based model:

$$\gamma(\phi) = (a \, \cos(\phi - \phi_0) - b)^+ \tag{9}$$

With $\gamma$ being response of the model neuron, $\phi$ is the stimulus' phase and $\phi_0$ the phase at which the response is maximal. Finally, $(\,)^+$ indicates an half-rectified function that only output zero or positive values. In this case $\chi = \frac{b}{|a|}$ and the modulation ratio can be obtained through the nonlinear mapping (see [39] for details):

$$\frac{F_1}{F_0} = g(\chi) = \begin{cases} \dfrac{-\chi\sqrt{1 - \chi^2} + \arccos \chi}{\sqrt{1 - \chi^2} - \chi \, \arccos \chi}, & \text{if } -1 \leq \chi \leq 1. \\[2ex] -\dfrac{1}{\chi}, & \text{if } \chi < -1. \\[2ex] \text{undefined}, & \text{if } \chi > 1. \end{cases} \tag{10}$$

In [39], authors further suggested that the classification of V1 cells as simple or complex might be caused uniquely by the nonlinear relationship between $\frac{F_1}{F_0}$ and $\chi$ rather than reflecting an actual neuro-physiological difference. Here, for simplicity, we refer to the same values conventionally used in literature: we consider model neurons as simple-like if $\frac{F_1}{F_0} > 1$ and as complex-like if $\frac{F_1}{F_0} \leq 1$. We use the modulation ratio also to evaluate the response to rotating stimuli in order to compare it to the response in the case of drifting phase. Although this measure is not in standard use to evaluate orientation tuning, it can be seen as a spectral analysis of the orientation response, similar to the work of Wörgötter and Eysel [56]. Nevertheless, to perform statistical tests on the different conditions, we analyze the distribution of $\chi$ (see S1 Text). Indeed, the $\chi$ index is easier to analyze as it does not show the typical bimodal distribution of $\frac{F_1}{F_0}$ and is linked to $\frac{F_1}{F_0}$ through a nonlinear, monotonous, and invertible function (Eq 10).

## Local homogeneity index (LHI)

To evaluate the functional implication of the orientation maps learned by our model, we used the local homogeneity index (LHI) as defined by [40]. The LHI measures the similarity of the preferred orientation in neighboring cells:

$$LHI(m) = \left| \frac{1}{k} \sum_n \exp\left( -\frac{(n - m)^2}{2\sigma^2} \right) \exp\left( 2j\theta_n \right) \right| \tag{11}$$

With $m$ being a location on the orientation map, $n$ a set of neighboring locations and $\theta_n$ the preferred orientation at $n$. Then, $j$ is the imaginary unit, so that $j = \sqrt{-1}$ and $|\,|$ represents the module of a complex number. The constant $k$ normalizes the measure and changes if $m$ and $n$ are bi-dimensional or mono-dimensional, for example in the case of MaxPool $2D_F$ or MaxPool $1D_F$. Finally, $\sigma$ is proportional to the width of the Gaussian window in which the LHI is calculated, here we set $\sigma = 1$ pixel. The LHI is bounded between 0 and 1, with high values corresponding to iso-orientation domains and low values to pinwheels. We define pinwheels as local minima in local neighborhoods of $3 \times 3$ pixels with LHI under a threshold of 0.2. In order to calculate the pinwheel density we define the average size of a cortical column as the ratio $c = \left( \frac{180°}{\text{mean(HWHH)}} \right)^2$, for each tested network. The pinwheel density is then defined as

the ratio:

$$\rho = \frac{\text{num. of pinwheels}}{\dfrac{M_s}{c}} \tag{12}$$

## Log-Gabor fitting

To estimate the optimal frequency and orientation for each model neuron, we fitted the synaptic weights of each cell with log-Gabor wavelets [31, 57]. This fit allows us to identify the preferred stimuli for each neuron in the first and second layers. Specifically, we can extract: the preferred orientation ($\theta$), phase ($\phi$), frequency ($f_0$) as well as the orientation tuning width (half-width at half-height, HWHH) [58]. Since second layer cells in the model project into the first layer of the network through a nonlinearity (pooling), we defined an approximated linear mapping $V_c^*$ for the second layer such that:

$$V_c^* = \underset{V_c}{\operatorname{argmin}} \frac{1}{2} \| x - V_c^T \gamma_c \|_2^2 \tag{13}$$

Specifically, $V_c^*$ is used to back project (and visualize) the second layer's weights in the input visual space. The kernels in $V_c^*$ were then fitted using log-Gabor functions to assess for the best orientation and spatial frequency to stimulate the second layer of the network.

## Drifting grating vs. Rotating grating

One goal of our study is to assess the ability of our network to predict complex behavior in V1, that is, to show phase-invariant responses. We test the network's invariance to drifting and rotating gratings. We do this to make sure that the networks we model indeed exhibit responses that are invariant to phase, yet that they remain tuned to orientation, as observed in neurophysiological experiments. Using the preferred orientation, $\theta$, and frequency, $f_0$, for each neuron in the first and second layer, we created sinusoidal grating with optimal parameters for each neuron. All stimuli were masked by a circular window of 14 pixels in diameter, that is, the size of the receptive field of model neurons. Finally, we evaluated the response of the network at different phases and orientations of the same grating (see Fig 5).

## Supporting information

**S1 Text. Analysis on $\chi$.**
(PDF)

**S1 Fig. Sparse neural activity maps. Left**. Example of synaptic weights learned from natural images $W_s$ and $W_c$, for the first and second layer, respectively. Here we show $V_c$, a linear approximation of $W_c$ (see section 'Drifting grating vs. Rotating grating'). Each kernel corresponds to a channel in the neural activity maps. **Right**. Representation of 3 channels from the neural activity maps ($\gamma_s$ and $\gamma_c$) elicited by the input $x$. Here, each pixel represents a model neuron and the color code indicates the amplitude of the neural response (lighter for no response and darker blue for the maximal response, here normalized to 1). The kernel in the top-left corner indicates the preferred stimulus of each channel.
(TIFF)

**S2 Fig. Emergence of topographic maps during learning.** We show the evolution of the topographic organization learned by $W_s$ during training when the SDPC network embeds the

MaxPool $2D_S + 2D_F$ function. Here, for $M_s = 100$. The weights are initialized to random values and the network gradually learns from input data. At first, we observe the emergence of edge detectors similar to the ones observed in the V1 of mammals. Gradually, thanks to the combined action of the feedback coming from the second layer of the network and the pooling function in the forward stream, neighboring cells in the topography become tuned to stimuli of similar orientations but different phases, generating a topographically organized map. (TIFF)

**S3 Fig. Complex cells population as a function of the network size.** We analyze the same results of Fig 5 in terms of the unimodal variable $\chi$ as defined in [39] (see section 'Modulation ratio and complex behavior'). In the **top-left** graph we illustrate the nonlinear relationship between $\chi$ and $\frac{F_1}{F_0}$. Here the results for $M_c = 100$ are shown. To avoid the assumption of normally distributed variables, we represent $\chi$ using the *median* $\pm$ *MAD* (median absolute deviation). Black ($\chi_\phi$) and green ($\chi_\theta$) lines indicate distributions obtained using drifting and rotating grating, respectively. The dashed lines indicate the value $\chi = -1$ for which $\frac{F_1}{F_0} = 1$, the threshold value for which a V1 cell is typically considered either simple or complex. Using drifting gratings as stimuli significantly generated lower values of $\chi_\phi$ (more complex-like cells) than rotating gratings, in all tested settings (one-tailed Wilcoxon signed-rank test). This result confirms that the network's invariance is linked to the stimulus' phase and that model cells remain narrowly tuned to orientation. For $p_s = $ MaxPool $2D_S$ and $p_s = $ MaxPool $2D_S + 1D_F$ the distribution of $\chi_\theta$ and $\chi_\phi$ do not vary significantly as a function of the network size. When the network shows a functional topographic map, for $p_s = $ MaxPool $2D_S + 2D_F$, we can see a clear dependency between $\chi_\phi$ and the number of features in the first layer of the network $M_s$. (TIFF)

## Author Contributions

**Conceptualization:** Victor Boutin, Angelo Franciosini, Frédéric Chavane, Laurent U. Perrinet.

**Data curation:** Victor Boutin, Angelo Franciosini.

**Formal analysis:** Victor Boutin, Angelo Franciosini.

**Funding acquisition:** Frédéric Chavane, Laurent U. Perrinet.

**Investigation:** Victor Boutin, Angelo Franciosini.

**Methodology:** Victor Boutin, Angelo Franciosini.

**Software:** Victor Boutin, Angelo Franciosini.

**Supervision:** Frédéric Chavane, Laurent U. Perrinet.

**Visualization:** Victor Boutin, Angelo Franciosini.

**Writing – original draft:** Victor Boutin, Angelo Franciosini.

**Writing – review & editing:** Victor Boutin, Frédéric Chavane, Laurent U. Perrinet.

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
