## [Decision Letter · Decision Letter 0]

20 Nov 2021

Dear Mr Angelo,

Thank you very much for submitting your manuscript "Pooling in a predictive model of early visual processing explains functional and structural diversity in V1 across species." for consideration at PLOS Computational Biology.

As with all papers reviewed by the journal, your manuscript was reviewed by members of the editorial board and by several independent reviewers. In light of the reviews (below this email), we would like to invite the resubmission of a significantly-revised version that takes into account the reviewers' comments.

We cannot make any decision about publication until we have seen the revised manuscript and your response to the reviewers' comments. Your revised manuscript is also likely to be sent to reviewers for further evaluation.

Sincerely,

Saad Jbabdi

Associate Editor

PLOS Computational Biology

Wolfgang Einhäuser

Deputy Editor

PLOS Computational Biology

Reviewer's Responses to Questions

**Comments to the Authors:**

Reviewer #1: Based on (partially their own) previous work, the authors extend the predictive coding framework in a hierarchical network setting towards non-linear (max) pooling. They show that simple and complex cells can emerge in the first and second layer, respectively, of a two-layer network. Moreover, depending on whether the pooling is over adjacent units in retinotopic space or in feature space, the simple cells are arranged in a salt and pepper, or in smooth orientation map-type layout, respectively.

The paper for most parts is in good shape and the formalism and implementation seem sound. The main results seem interesting and important and I would like to understand them better. However, the explanations lack clarity at times. Below I point out some of the parts where I got stuck, hoping the authors can improve those parts, as this would surely facilitate the impact of their ideas.

Fig. 3 and text: In case there is a smooth orientation map, do also complex cells form such a map? With preferred orientations matched to those of the simple cells?

3.2 along with Fig. 3: What exactly is shown in the left column and how does this link to Fig. 1? Is it all layers, or a subset, or just one layer? If several layers, are the neurons whose weights are shown all located at the same position across layers?

3.1.: Here the authors say convergence is the first important result, but then in the conclusions they argue that the problem is convex and convergence therefore guaranteed?

Please explain better p_i^-1 after Eq. 5.

Fig. 1: what does the cross in the circle mean? Why is the black down-pointing error targeting the neuron layers, and not a point between the Relu and p(gamma)? What would ‘a’ correspond to in this fig?

Eq. 9: I don’t see the Hebb rule. Please make this more obvious.

4.31 “..it only depends on the type of pooling function used.” I suppose, it does also depend on the input data, or not?

The results seem interesting, but I still remain a bit confused when trying to appreciate the proposed mechanism for map formation. Could the author please provide a better intuition here? Why does max pooling in layer 2 across nearby feature channels give rise to a continuous variation in the preferred orientation across retinotopic space (in layer 2, I suppose)?

Reviewer #2: The paper elaborates on the authors' earlier work on hierarchical sparse predictive coding, but here incorporating a pooling mechanism between layers. The model is shown to account for the structure of orientation maps in V1, and provides an account that ties this structure together with complex cells.

Overall I find this paper very interesting. The model is novel and potentially has interesting properties that go beyond previous work. My main concern is the suitability for PLOS, as it is currently written. The computational neuroscience literature is by now fully saturated with models that claim to account for orientation maps and pinwheels in various ways. Against this backdrop, it is not very clear what this paper contributes to the story. On the other hand, it seems the motivation from the introduction is about invariance, and how to tie this idea into the sparse predictive coding framework, which is a very rich and interesting question begging for a good solution. The main result though is about orientation maps and pinwheels, so I am left scratching my head.

I would encourage the authors to rethink the narrative of this paper. It also needs to be shortened. The writing in the intro and discussion is at times long-winded. It is difficult to follow the thread, and the main point of the paper and its central contribution gets lost. Focus on what is missing from the orientation map/pinwheel story, and how does this paper fill the gap? (assuming that is the story) If the story is about invariance and complex cells, then focus on what is missing from that and how his paper contributes. Or if the story is about the relation between the two, then make that the focus and explain it clearly.

Specific comments:

This work seems very similar in spirit to Hyvarinen & Hoyer's Topographic ICA, as the authors point out. Some of the differences are discussed, but I am having a hard time seeing a fundamental distinction, at least as it relates to orientation maps and pinwheels. In my mind that model does a remarkable job explaining these phenomena.

Using a convolutional neural network seems implausible as a neurobiological model. In a convolutional model you are dictating the stride of the filters - i.e., how they tile space. By contrast in a fully connected model, the system must figure out not only how to tile space, but all the other feature dimensions such as orientation, spatial frequency, color, disparity, motion, etc. Presumably biology had to confront a similar problem, and the solution is what we see is in the various maps found in V1. Perhaps I do not fully understand how the convolutional model is being utilized, but I think the authors need to do a better job justifying this choice, as it seems to presume part of the structure that you are attempting to explain in the biology.

The pooling operation is introduced in Section 2.1, but not defined until Section 2.3. I was very confused until I finally got there. And then, I don't understand why max pooling. Part of the justification seems to be that it is popular, but that is sociology. The other argument refers to winner take all, but is that right? Sparse coding already induces a competition - a kind of multiple winner take all. So if that is already happening, why do you need the extra max step in the pooling function? this did not seem well motivated.

Also, wouldn't you want to learn the pooling function? Isn't there a way to incorporate that into your learning rule?

Back to Section 2.1, the definition of g_i in eq. 3 comes before how it is used in eq. 4. I would recommend swapping the order of these.

In Figure 1, panel b, the filters shown in the different layers of the convnet - along the M_s dimension - are all the same. Shouldn't they be different? this causes confusion.

You p^-1 to denote the derivative of p and refer to it as an "unpooling" function. Is that right? I am confused by this. Is it because you can't take the derivative of p since it is a max pooling function? how would such a function be implemented?

The second layer weights W_c - are they learned as well? what do they look like? I didn't seem them shown in any of the figures. It seems like this is a potentially very interesting aspect of the model that is not brought out.

Other comments/impressions:

Overall what interests me about this paper the most is the idea of incorporating invariance into a sparse predictive coding hierarchical model. So I am surprised that aspect is almost played down, and the paper instead focuses on orientation maps and pinwheels. Yes some animals have them, some don't. Presumably there is some topographic organization the system is pooling over, but that may not be overt when you look at the cortex. Just because cortex is organized as a two-dimensional sheet does not imply that we should restrict ourselves to 2D topographic models. For example the intrinsic structure may 4D, but packed into a 2D sheet and using the thickness of that sheet to interweave the other dimensions. It would seem to me that one of the more interesting predictions of this model would be to show what the W_c weights look like, and illustrate how the top-down connections from the second layer influence the first layer. That is truly novel, and there are few models that make concrete statements about this in terms of a normative representation principle. BTW hierarchical sparse coding was also the basis of the famous first deep network model out of Google brain that learned "cat neurons," which used a kind of topographic ICA, see https://ieeexplore.ieee.org/abstract/document/6639343 - although they did not have any feedback since it was not formulated as a generative model. Yours is, which makes it potentially more interesting.

Some other related works to consider:

The sparse manifold transform of Chen et al. (2018), and this SFN 2019 poster:

"A geometric theory for complex cells" https://www.abstractsonline.com/pp8/#!/7883/presentation/44286

provide another interpretation of pooling as part of a transformation to discover the underlying geometry in a sparse code. See also this related paper by Hosoya & Hyvarinen on learning the pooling function:

Hosoya, H., & Hyvärinen, A. (2016). Learning visual spatial pooling by strong PCA dimension reduction. Neural computation, 28(7), 1249-1264.

Finally, there is this related work by Osindero & Hinton

Osindero, S., Welling, M., & Hinton, G. E. (2006). Topographic product models applied to natural scene statistics. Neural Computation, 18(2), 381-414.

Spelling/typos:

therm  term

**Have the authors made all data and (if applicable) computational code underlying the findings in their manuscript fully available?**

Reviewer #1: None

Reviewer #2: Yes

PLOS authors have the option to publish the peer review history of their article (what does this mean?). If published, this will include your full peer review and any attached files.

Reviewer #1: No

Reviewer #2: **Yes: **Bruno A. Olshausen
---

## [Decision Letter · Decision Letter 1]

1 Jun 2022

Dear Mr Boutin,

We are pleased to inform you that your manuscript 'Pooling strategies in V1 can account for the functional and structural diversity across species.' has been provisionally accepted for publication in PLOS Computational Biology.

Please note the reviewer's remaining comments, which may help improve the clarity of the paper. 

Best regards,

Saad Jbabdi

Associate Editor

PLOS Computational Biology

Wolfgang Einhäuser

Deputy Editor

PLOS Computational Biology

Reviewer's Responses to Questions

**Comments to the Authors:**

Reviewer #1: The authors have addressed most of my previous concerns and the manuscript has improved in clarity. The effect seen in Fig. 3a is still a bit confusing to me. This figure part is never really explained anywhere in detail. What is the intuition? In the text, Fig. 3a is mentioned only briefly, and only after the figure parts b-d were discussed, which seems to be a strange ordering.

Reviewer #2: The paper has been improved, but there remain many points of confusion. I would encourage the authors to put more work into making the paper clear, as is it seems rather sloppily put together, and that will not encourage the reader to take the work seriously.

The equations and description of the model appear under the "Results" section - this seems an odd choice, because these things are not results, they are the authors specification of the model. So perhaps put it in its own section called "Model" or something like that.

Equation 1 specifies a general N layer architecture, and then it is stated that for the rest of the paper N=2. Equation 2 uses the notation W_1 and W_2 etc. which is consistent with that. But then eqs. 3 and 4 slip back to the general N layer notation, which is confusing. Pick one and stick with it.

"max-pooling is not derivable"  "max-pooling is not differentiable"

Figure 1 panel (a) refers to the W_2 layer (here labeled W_c - note more confusion) as "complex cells." But it would seem that the output of the pooling units that feed into the W_2 layer would constitute what most people would consider complex cells. The W_2 layer which takes linear combinations of these would correspond more to "hypercomplex" cells (as hubel and wiesel called them). This whole treatment I find rather confusing and sloppy.

Regarding the topographic organization - which seems to be one of the main points of the paper - it appears there are two forms to this organization: 1) the organization along the feature dimension, for example arranging pooling into sqrt(M)xsqrt(M) blocks, and 2) the pooling over the stride of the convolution, which by definition is over the spatial dimension. If you were a neurophysiologist probing this system, how would you know which one you are looking at? does it matter? This setup seems rather complicated and contrived to me.

Overall I think the authors are doing something interesting here, but the presentation is messy and complicated and difficult to read. I would encourage the authors to put more work into making these things clear.

**Have the authors made all data and (if applicable) computational code underlying the findings in their manuscript fully available?**

Reviewer #1: Yes

Reviewer #2: **No: **

PLOS authors have the option to publish the peer review history of their article (what does this mean?). If published, this will include your full peer review and any attached files.

Reviewer #1: No

Reviewer #2: No

---

## [Editor Report · Acceptance letter]

18 Jul 2022

PCOMPBIOL-D-21-01332R1 

Pooling strategies in V1 can account for the functional and structural diversity across species.

Dear Dr Boutin,

I am pleased to inform you that your manuscript has been formally accepted for publication in PLOS Computational Biology. Your manuscript is now with our production department and you will be notified of the publication date in due course.

With kind regards,

Agnes Pap
